# Biopolymeric Fibrous Aerogels: The Sustainable Alternative for Water Remediation

**DOI:** 10.3390/polym15020262

**Published:** 2023-01-04

**Authors:** Alejandra Romero-Montero, José Luis Valencia-Bermúdez, Samuel A. Rosas-Meléndez, Israel Núñez-Tapia, María Cristina Piña-Barba, Gerardo Leyva-Gómez, María Luisa Del Prado-Audelo

**Affiliations:** 1Departamento de Farmacia, Facultad de Química, Universidad Nacional Autónoma de México, Ciudad de México 04510, Mexico; 2Tecnológico de Monterrey, Escuela de Ingeniería y Ciencias, Campus Ciudad de México, Ciudad de México 04510, Mexico; 3Tecnológico de Monterrey, Escuela de Ingeniería y Ciencias, Campus Santa Fe, Ciudad de México 04510, Mexico; 4Instituto de Investigaciones en Materiales, Universidad Nacional Autónoma de México, Ciudad de México 04510, Mexico

**Keywords:** aerogels, biopolymers, water remediation

## Abstract

The increment in water pollution due to the massive development in the industrial sector is a worldwide concern due to its impact on the environment and human health. Therefore, the development of new and sustainable alternatives for water remediation is needed. In this context, aerogels present high porosity, low density, and a remarkable adsorption capacity, making them candidates for remediation applications demonstrating high efficiency in removing pollutants from the air, soil, and water. Specifically, polymer-based aerogels could be modified in their high surface area to integrate functional groups, decrease their hydrophilicity, or increase their lipophilicity, among other variations, expanding and enhancing their efficiency as adsorbents for the removal of various pollutants in water. The aerogels based on natural polymers such as cellulose, chitosan, or alginate processed by different techniques presented high adsorption capacities, efficacy in oil/water separation and dye removal, and excellent recyclability after several cycles. Although there are different reviews based on aerogels, this work gives an overview of just the natural biopolymers employed to elaborate aerogels as an eco-friendly and renewable alternative. In addition, here we show the synthesis methods and applications in water cleaning from pollutants such as dyes, oil, and pharmaceuticals, providing novel information for the future development of biopolymeric-based aerogel.

## 1. Introduction

Water pollution is one of the most relevant current ecological problems since, due to industrial and human activity, water bodies and landfills have become contaminated with toxic substances such as oils, dyes, pharmaceuticals, and heavy metals [1]. These substances are a public health issue and cause problems such as blockage of water purification plants, damage to aquatic organisms, and an imbalance in environmental bacteria. Additionally, the dispersibility and infiltration of water facilitate the distribution of various contaminants in other natural sources and therefore represent an additional challenge in remediation [2]. Consequently, it is in our interest to reincorporate the remediation materials into the environment through the treatment of water with biopolymeric fibrous aerogels. Due to the properties of these components, several alternatives have been proposed to remove them from water, such as separation membranes, physicochemical purification processes, chemical deposition, and, recently, aerogels for adsorption, filtration, and remotion of contaminants [3]. These properties of aerogels can aid in rapidly adsorbing pollutants from water and allowing water reuse, thus emerging as an effective, economical, and environmentally friendly alternative.

Aerogels are three-dimensional solid materials with high porosity and surface area that come from organic and inorganic precursors or mixtures [4]. Aerogels were first synthesized in 1931 by Professor Kistler [5] and, in recent decades, have gained relevance due to the diversity of potential applications they could have because of their versatile chemical structure [6]. One of the great advantages of these materials is that their properties can be modulated depending on the application to be achieved. The precursor’s selection can define the final product’s characteristics; for example, cellulose-based aerogels have a strong hydrophilic character and low lipophilicity [7,8,9], while silica-based aerogels are excellent thermal insulators [10].

Several methodologies have been implemented to improve the final characteristics of the aerogel, such as chemical modification of the surface, the inclusion of additives, or the encapsulation of bioactive molecules. One of the most used strategies is the addition of nanoparticles that confer different capabilities to the material, such as magnetism [11], capacitance [12], or better mechanical properties [13]. Finally, it is important to know that the determining step in the success of aerogel synthesis is obtaining a sufficient pore without it collapsing or losing its structural characteristics during the process [14].

Following these characteristics, aerogels have been studied from various perspectives, including as materials for tissue regeneration, thermal insulators, and bioremediation. In recent years, it has been demonstrated that they have high efficiency in the removal of pollutants from the air [15], soil [16], and water [17], which promotes them as a promising alternative to use as a remediation method.

The use of biopolymeric aerogels is highly attractive with the intention of preserving and favoring remediation processes [18]. For example, to extract biopolymers in high quantities, it is possible to use natural sources that perish [19] as biomass, with the great advantage of their reincorporation into the environment after several reuse cycles. Additionally, modifications to natural polymers can be guided to maintain biodegradation processes and not incur additional contamination, especially with the addition of synthetic grafts.

Cellulose and chitosan [20] are two biopolymers widely explored in materials engineering. They are two materials that constitute a support system for plants and crustaceans and currently predominate as waste at an industrial level. There are widely explored methods of obtaining and purifying materials that allow adequate reproducibility in different applications. Similarly, it is possible to know different modification routes on the monomers that facilitate applications in hydrophobicity, porosity, mechanical resistance, and biodegradation. Other biopolymers of industrial interest in biopolymeric aerogels include cotton [21] and lignin [22].

Additionally, the new trends in aerogels with biopolymers prefer that the new biopolymers have the capacity for several reuse cycles, exhibit high specificity, and maintain biodegradation capacity. A challenge that requires innovative materials engineering [21].

The purpose of this article is to describe and analyze the state of the art of biopolymer-based aerogel manufacturing methods and their novel applications in water bioremediation, as well as provide perspectives for their use.

## 2. Biopolymers Employed

In the last decade, the generation of aerogels based on biopolymers has attracted the scientific community’s interest due to the need to generate materials that have a sustainable origin and come from cleaner and more environmentally friendly processes [23,24]. Such interest has grown exponentially in recent years, as evidenced by the increase in publications on the subject [25], showing that the most researched biopolymers so far are polysaccharides (i.e., cellulose and its derivatives, alginates, chitosan, pectin, and starch), proteins [26,27], and to a lesser extent phenolic compounds (i.e., tannins, lignin). These biopolymers have attracted attention because they represent a sustainable option compared to traditional crude oil-derived polymer precursors and versatile chemical structures with an abundance of potentially modifiable functional groups. This is attractive from the point of view of generating selective materials with controllable structures and modulable physicochemical properties [28,29]. In addition, these molecules are innocuous, non-toxic, biocompatible, and easily assimilated by living beings as they are part of the normal diet. Furthermore, from the industrial point of view, these materials can be obtained on a large scale economically due to the abundance and availability of these biopolymers, to the extent of becoming an alternative for the use of waste from large food and beverage industries [14,30].

Many materials have been used in a pure state or combined with other crosslinkers to fabricate aerogels. Each material’s physical, chemical, mechanical, and biological properties eventually determine its desired aerogel properties.

### 2.1. Cellulose

Cellulose is the most abundant and widely used biopolymer due to its large surface area, high mechanical strength, and tunable chemical surface. Cellulose is a linear polymer formed by the linkage of D-glucose with 1,4-glycosidic bonds. Further, each glucose repeat unit in cellulose has three hydroxyl groups with high chemical reactivity, which makes the use of crosslinking agents unnecessary in the aerogel preparation process. Moreover, the intramolecular and intermolecular hydrogen bonds allow for physical crosslinking, which generates stable three-dimensional structures [31,32].

The chemical structure also makes cellulose susceptible to chemical modifications that improve its mechanical and structural properties. Additionally, depending on the source (plant or bacterial) and the extraction method, the polymer can have different lengths of the chain (degree of polymerization), size, degree of crystallinity, and thermal stability [33]. Cellulose extracted from plants (rice, bagasse, cotton, wood, etc.) contains impurities such as lignin and hemicellulose that affect the properties of the resulting material [34]. In contrast, cellulose obtained from *Acetobacter xylinum* bacterial cultures is purer and has a higher percentage of crystallinity [35].

These features allow tuning its interaction with other polymers, nanoparticles, and small molecules [36]. The mechanical properties, pore size, and adsorption capacity of cellulose-derived materials depend largely on their crystalline state and processing method [31,37]. For example, to improve the solubility of cellulose, lignin, and soybean meal aerogels and to functionalize them with bioactive molecules such as tannins, the sulfation process, and supercritical CO_2_ drying techniques have been used with excellent results; these aerogels have potential applications as anticoagulant agents and also in bioremediation due to their surface chemical characteristics, low density, and high porosity [38,39,40].

### 2.2. Chitosan

Chitosan is obtained through the deacetylation of chitin, a linear amino polysaccharide formed by N-acetylglucosamine units connected through β-(1,4) bonds. The degree of deacetylation of chitosan is greater than 50%, and it is highly soluble in dilute acids (pH < 6) [41]. Logically, being a natural polymer, the chain size and all its physicochemical properties depend on the source (insect and mollusk shells) and the extraction method. Chemically, the amino groups confer unique properties as they are protonated increasing NH_3_^+^ solubility and allowing the formation of nanostructures due to intermolecular interactions, which exhibit biological functions as coagulants for wastewater [42]. The primary amine groups display a high reactivity in nucleophilic addition to aldehydes, followed by dehydration to form imine bonds (Schiff’s bases). This allows it to be chemically functionalized with molecules that improve its mechanical and structural properties [41,43].

Chitosan and alginate are generally selected for the presence of amino and carboxyl functional groups, which can be easily modified by solvothermal methods, obtaining different structures depending on the pH of the medium [44]. Chitosan is not soluble in water, allowing them to preserve its structure in aqueous media and increasing the scope of its applications. The aerogels thus obtained are generally mesoporous and anisotropic [45,46]. These materials reach particularly high degrees of polymerization and crystallinity.

### 2.3. Alginate

Alginate is a polysaccharide extracted from seaweed that contains α-l-guluronic acid and β-D-mannuronic acid residues, which are linearly linked by a 1,4-glycosidic bond. The presence of carboxylate groups within G units confers a global negative charge at pH = 7, which is usually compensated by using Na+ cations as counterions. Gelation of alginate takes place by inducing the cross-linking of the alginate polymers with divalent cations (usually Ca^2+^) following the “egg-box” gelation model mechanism [47]. Alginates aerogels are characterized by unique properties, such as a large surface, open porosity, good compatibility and biodegradability, low thermal conductivity, a high potential for insulation applications, good flexibility, and are classified as viscoplastic materials. In addition, by adding Ca^2+^ or Al^3+^ ions, it’s possible to improve their mechanical properties [48,49].

### 2.4. Others

In other cases, biopolymers are selected to substitute non-renewable precursors, i.e., replacing polyol with cellulosic derivatives and plant oils; tannin and lignin polymers often substitute resorcinol and copolymerize with formaldehyde to form resin [50,51]. For example, in pectin (a diverse group of anhydrogalacturonic acid polymers) and starch with a range of methyl esterification of the carboxyl groups (degree of esterification, DE) [52,53], the microstructure and final performance of the gels depend on the chemical structure of the polymer sources (mainly DE) and the gelation mechanism [54]. Bovine serum albumin and oligo- and polypeptide nanofibrils are generally used to obtain protein aerogels. The generation of these materials is controlled at the supramolecular level. It depends on self-assembly interactions between peptide and protein molecules to form the desired three-dimensional structure, generally dominated by the interaction of phenylalanine or leucine units [55,56].

A suitable combination of materials can exhibit superior properties and new functionalities compared to their single-component counterparts. Biopolymer additives can reinforce the inherent brittle structure of inorganic aerogels or introduce multifunctionality. Moreover, biopolymer aerogels’ intrinsic density, strength, deformation, flammability, and hygroscopicity can be improved by adding inorganic components to biopolymer aerogels [57,58].

## 3. Preparation Methods

The methods for obtaining bio-aerogels start with the formation of a mixture of precursors reduced to their molecular form or nanostructures (chitin nanofibrils, nanofibrous cellulose, or protein aggregates), followed by a gelation step where the concentration and functional groups of the precursor, the pH, and the concentration and type of crosslinker are crucial parameters [59]. The key final phase is the removal of the solvent from the pores of the wet gel without modifying the pore size and shape. It has become necessary to adapt their preparation to modulate the final characteristics of the material, such as three-dimensional pore structure, surface functionality, and adsorption capacity [60,61]. In the following sections, we will present some of the most common and efficient methods for obtaining bio-based aerogels, according to the most recent reports (Figure 1).

### 3.1. Supercritical Carbon Dioxide Drying

The most common method of obtaining aerogels is drying them with CO_2_ in a critical state (304 K, 7.4 MPa) due to its low cost and high safety. This method involves a two-way mass transfer as scCO_2_ replaces the solvent (usually ethanol, methanol, amyl acetate, or acetone) inside the pores of the wet gel. Then the fluid mixture inside the pores reaches supercritical conditions without any vapor-liquid transition. The absence of liquid phases within the pores allows for no surface tension, preventing pore collapse with solvent removal [62].

The presence of water within the wet gel network can cause damage to the resulting aerogel structure due to its high surface tension; therefore, aerogels that come from a synthesis method that involves having water remaining (such as those synthesized in ionic liquids) must undergo a water-acetone or water-ethanol solvent exchange for the successful preservation of the three-dimensional structure [63,64]. The solvent exchange process by soaking is very slow and generally takes more than 48 h, but it is a critical step for successful drying. The drying process with scCO_2_ helps to reduce the risk of pore collapse due to capillary pressure in the pores, generating materials with a homogeneous structure [65,66]. Supercritical drying can use other fluids such as NO_2_, xenon, etc. and is also known as critical-point drying or supercritical lyophilization. This inexpensive process is nevertheless expensive in terms of infrastructure due to the high-pressure equipment required.

### 3.2. Freeze-Drying

Freeze-drying is a simple and environmentally friendly method based on removing solvent from the gel by sublimation after freezing the gel, allowing dehydration to occur at a low temperature. Ideally, the solvent should be water so that aerogel production can be carried out on a large scale at the lowest possible cost. In this process, no liquid-gas interface is developed, so there is no structural collapse caused by capillary force [59]. Typically, the process consists of freezing the wet gel (−45 °C to −15 °C) before starting the sublimation process under vacuum conditions [67].

Furthermore, the water crystals thus formed serve as templates for creating the pores and are removed in the first drying phase. In contrast, the remaining water that could be bound or adsorbed on the surface of the pores is removed in the second drying step [68]. It is known that temperature, freezing time and speed, ionic strength, and vacuum pressure are factors determining the final properties and morphology of the aerogel. For example, high freezing temperatures increase the porosity but at the same time favor the denaturation of the biopolymers and, with them, the degeneration of the aerogel [69,70].

### 3.3. Ambient Drying

The methods described above are difficult to scale up because batch processing is slow and involves the use of expensive equipment that must be operated at high pressures and temperatures; therefore, drying aerogels at room temperature has recently become relevant, a technique that was first described in 1995 by Prakash et al. [71]. The success of this technique is dependent on the fact that the three-dimensional structure of the aerogel is reinforced before drying, which is achieved through the reduction of capillary pressure by surface modification and solvent exchange; it is also known that cross-linking reactions must be eliminated [72].

As already mentioned, during the drying of aerogels, there is an attractive stress on the liquid inside the gel pores, which causes a contraction (compressive stress in the solid phase), followed by a continuous cross-linking reaction of the gel network, resulting in many cases in an irreversible shrinkage. The gel remains in its most compact state and reduces its porosity by 10–60% [73]. Supercritical drying eliminates the vapor-liquid interface; however, the solid-liquid interface still affects the material obtained. Therefore, ambient drying proposes, in addition to rapid solvent exchange, a chemical modification of the gel surface with molecules that are not susceptible to forming hydrogen bonds or condensation reactions (i.e., organosilanes, TEMPO, etc.), eliminating the possibility of the structure collapsing due to capillary tension and permanent shrinkage due to crosslinking [74].

## 4. Aerogel Application in Water Remediation

Increasing industrialization comes with growing pollution of water bodies by contaminants such as oil [75], salts, dyes [76], or pharmaceuticals [76,77], among others (Figure 2). To remediate water, separation techniques include air flotation, coagulation, chemical separation, biological treatment, and membrane separation technologies [24,75]. This section will discuss the use of biopolymeric aerogels to treat water from different pollutants.

### 4.1. Oil Recovery

Oil is considered one of the most dangerous pollutants in water, causing serious environmental problems [9,78]. Recently, aerogels have been studied as oil/water separation methods due to their low density, high porosity, hydrophobicity, and large surface area [79]. Carbon nanostructured aerogels such as carbon nanotubes or graphene have attracted much attention. Further, due to harmful precursors and other complications [80], environmentally friendly and biodegradable materials such as carbon-based biomass are preferred [81].

Water has been treated against diverse pollutants, such as organic solvents and oils, with different biopolymers. Table 1 presents a collection of aerogels synthesized from various raw materials and hydrophobic modifications. The results depicted for each raw material treated with a hydrophobic modification are the porosity of the sample, the wet contact angle (WCA), the absorption capacity for the pollutants treated, the oil recovery methods, the reusability of the sample, and the regeneration percent on the last cycle.

Additionally, the aerogels must be highly porous as well as hydrophobic and oleophilic at the same time to ensure the selectivity of oil absorbance against water [82]. By measuring the water contact angle (WCA), a material is classified as hydrophilic (WCA < 90°), hydrophobic (90° < WCA < 150°), or superhydrophobic (WCA > 150°) [78]. Most samples have large porosity (>87%) and are hydrophobic, reporting WCA between 107.2–146.5°.

Further, the surface of biopolymeric aerogels can be functionalized with methods such as a silylation routine to increase the hydrophobicity of the sample. Silylation alters the microstructure by switching it from regular porous cells to a random tridimensional assembly of thin sheets [83]. The samples without hydrophobic modification exhibit WCA between 132–145° [80,81,84,85,86,87]. The surface silylation could be achieved using methyltrimethoxysilane (MTMS) [9,24,83,84,85,86,87,88,89,90,91] (WCA 130–152°), trimethylchlorosilane (TMCS) [82] (WCA 137–146°), hexadecyltrimethoxysilane (HDTMS) in a combination of MTMS [78] (WCA 107–159), or silicon ceramic coating powder (SCCP) [92].

Recently, Jiang et al. [24] synthesized a super-hydrophobic aerogel by combining the properties of lignin with sodium alginate (SA) and graphene oxide (GO), achieving silylation growth and obtaining a WCA of 161°. This 3D aerogel possessed a porous structure and low density and presented a switching ability from the absorption of the pollutant (oil or organic solvents) to desorption with a change in medium pH.

Similarly, cellulose aerogels from different sources employed for oil/water separation have been reported. In 2015, Sai et al. [82] obtained a hydrophobic bacterial cellulose aerogel with porosity as high as 99.6% by modifying the surface of the bacterial cellulose aerogel with a TMCS reaction. The result was an increase in the WCA, reaching a value as high as 146.5°. The absorption capacity (g/g) is defined as the mass of absorbed oil per mass of dry absorbent, which is necessary to prove the effectivity of the aerogel [81]. Absorption capacity ranges between 80–175 g/g in different oils and solvents for hydrophobic bacterial cellulose aerogels. As discussed by Sai et al., absorption capacity is directly proportional to the pollutant’s density [82]. Low-density pollutants such as acetone and gasoline have low absorption capacities, while chloroform usually has the highest.

On the other hand, one of the main characteristics of using aerogels is that the pollutant is recovered through different methods, including burning, distillation, solvent extraction, vacuum filtration, or mechanical compression (squeezing) [9]. The last one is the most used since it is simple and environmentally friendly. It does not use thermal energy, such as burning and distillation, or chemical solvents, such as extraction. Reusability is measured by recovering the pollutant and repeating the absorption in various cycles. The stability of the absorption capacity is desired, but sometimes the extraction of residual solids decreases it. In the case of the bacterial cellulose aerogel [82], the reusability was tested by performing 10 cycles with diesel and maintaining 120 g/g.

Cellulose has also been tested in birch and waste sources. A study by Laitinen et al. [78] in 2017 compared industrially bleached birch kraft pulp and fluting board with recycled box board and milk-container board, producing aerogels by nanofibrillation, freeze-drying, and silylation. The fluting board showed to be super-hydrophobic (WCA = 159°). The results show remarkably high absorption rates of various solvents and oils (65–205 g/g) with birch and fluting boards performing better than the recycled ones. The reusability was performed with marine diesel oil, decreasing each cycle’s absorption capacity to 70–82% by the end of 30 cycles [84]. This report shows that municipal solid wastes may be used as a source of cellulosic material for aerogels, increasing their circular production. Another example of cellulose obtained from waste was provided by Han et al. [84], who used waste newspaper subject to freeze-drying and pyrolyzing. The scanning electron microscope (SEM) micrographs demonstrate porous networks for the waste newspaper fiber aerogel before the pyrolysis (WNP) and after the pyrolysis (CA). The former demonstrates to be hydrophilic, while the latter exhibits a WCA of 132°, even floating in the water. The absorption capacity is 29–51 g/g due to its porosity, which is high when compared with certain inorganic-based aerogels such as zeolites or graphite, although not as high when compared with carbon nanotubes or carbon nanofiber aerogels. Further, the absorption capacities remain stable after 5 cycles for gasoline (recovered by distillation due to its low boiling point) and ethanol (recovered by combustion due to its flammability). However, for pump oil (recovered by mechanical compression since it has a high boiling point), the capacity was reduced to 31.5% because there was an incomplete compression of the aerogel, so some pump oil remained in the sample.

Other biomaterials in producing oil recovery aerogels include winter melon, bamboo, cotton, pineapple, sisal, and sugarcane bagasse. In 2014, Li et al. [81] elaborated aerogels with winter melon, one of the first biomass-based aerogels to be recently reported. Their product generated an absorption capacity 16–50 times its weight with stable recyclability for almost all the pollutants assessed, except for crude oil, since certain solids such as asphalt were not removed by distillation.

Bamboo chopsticks were used by Yang et al. [80], reaching absorption rates as high as 129 times their weight. They used distillation, combustion, and squeezing to measure recyclability, arguing that the former is used for valuable pollutants or those with low boiling points. In contrast, the latter is used for high-boiling-point pollutants. The squeezing technique was used for simplicity, but it proved to be the least efficient due to incomplete compression. Yuan et al. [79] reported that bamboo pulp aerogels presented an extraordinary absorption rate of even 150 g/g and stable reusability.

Recently, in 2020, Yi et al. [91] synthesized a super-hydrophobic (WCA = 152.1°) aerogel with bamboo fungus using MTMS as a hydrophobic modifier (Figure 3(1)). Absorption capacity varies from 20–42 g/g as seen in Figure 3(2), with stable recyclability and reusability after 10 cycles.

The cotton and cotton/cellulose aerogels were compared by Cheng et al. [9] using MTMS as a hydrophobic modifier. The combined synergistic effect of both fibers (cotton and cellulose) demonstrated better results than cotton alone, with absorption capacities of 72.3 g/g in machine oil and 94.3 g/g for dichloromethane. In 2019, Zhang et al. [87] fabricated a superlight absorbent with cotton nanocellulose and sodium dodecyl sulfate (SDS). The result yielded absorption capacities as high as 207 g/g, although other parameters such as porosity, WCA, or recyclability were not reported.

Pineapple leaves, sisal leaves, and sugarcane bagasse with hydrophobic modifications have been reported in recent years. Do et al. [88] absorbed motor oil with pineapple, leaving waste modified with MTMS. The product shows an average WCA of 140° and an average absorption of up to 37.9 g/g. Additionally, sisal leaves were functionalized with Cu nanoparticles by Li et al. [93] in 2021, exhibiting superhydrophobicity (WCA = 150.3°) and high oil absorption capacity (67.8–164.5 g/g) with a very high separation efficiency (above 97%).

Another example of ultralight aerogel was provided by Kumar et al. [92] using waste sugarcane bagasse (SBA) modified with SCCP and elaborated by freeze-drying. Figure 4(1) shows the macroscopic structure that can stand on the leaf of a flower due to its very low density. Hydrophobic behavior is demonstrated in Figure 4(2) with a WCA of 140.1°. This aerogel uptakes 23 times as much crude oil compared to its weight, with excellent recyclability after 10 cycles (Figure 4(3)). The aerogel’s flexibility was related to its 3D porous network, as observed by electronic microscopy. The authors evaluated different sugarcane bagasse concentrations, finding that the increment in this component generates a decrement in the pore size and a collapse in the macropores and micropores, thereby decreasing the cumulative pore volume and pore size.

Despite cellulose, other biopolymer candidates to be employed as aerogels are lignin and chitosan. The most recent experiments with lignin as a source have yielded very interesting results. The ligning with polyvinyl alcohol composite aerogel and MTMS modification prepared by Yi et al. [90] led to an impressive absorption capacity of 1200 g/g with chloromethane with a separation efficiency of up to 94% and stable reusability. As discussed previously, a super-hydrophobic aerogel membrane was developed by Jiang et al. [24] by combining the features of graphene oxide (GO), sodium alginate (SA), lignin, and MTMS silylation, achieving 96.7% separation efficiency.

Finally, chitosan (CS) has been intensively studied due to its amine and hydroxyl groups, which act as active sites for pollutants [85]. Li et al. [85] obtained an aerogel that was proven for both organic pollutants and heavy metal ions. The absorption capacity was 13.11–32.39 g/g for organic pollutants [89]. An improved product was performed by Yi et al. [89] using chitosan from shrimp shells with MTMS modification. SEM images (Figure 5(1)) of the cross-sectional area of the pristine and modified aerogel indicate a highly porous network. The modified sample proved to be super-hydrophobic (152.3°), increasing the surface wettability compared to the pristine sample (Figure 5(2)). The absorption capacity (31–63 g/g) showed dependence on the density of the solvents, as perceived in Figure 5(3). When proven by 10 cycles of mechanical compression, the absorption efficiency remained stable.

**Table 1 polymers-15-00262-t001:** Aerogels are used for oil/water separation.

Raw Material	Hydrophobic Modification	Porosity (%)	WCA (°)	Pollutants Treated	Absorption Capacity (g/g)	Recovery Methods	Reusability (Cycles Proved)	Regeneration % Last Cycle (Substance)	Ref.
Bacterial cellulose hydrogel	TMCS	99.6	137.1–146.5	Inorganic oils (Gasoline, diesel, paraffin liquid) Organic oils (plant) Organic solvents (*n*-hexane, Acetone, Toluene, Chlorobenzene, Dichloromethane, Chloroform)	80–175	Mechanical compression	10	Stable (Diesel)	[82]
Bamboo	MTMS	99.4	140	Inorganic oils (Gasoline, diesel, pump, mineral, motor) Organic oils (corn) Organic solvents (Acetone, Ethanol, Toluene, Hexane, Chloroform, Dimethyl sulfoxide)	45–99	Mechanical compression	35	84 (gasoline)	[83]
Bamboo chopsticks	-	-	145	Inorganic oils (pump, crude, diesel, gasoline) Organic oils (colza) Organic solvents (Hexane, Octane, Decane, Hexadecane, Chloroform, Toluene)	30–130	Distillation, combustion, mechanical compression	6	Stable (n-Hexane distillation) 62.1 (Hexadecane combustion) 61.4 (gasoline compression)	[80]
Bamboo fungus	MTMS	93.3	152.1	Oils (Corn, Silicone, Gasoline, vaccuum pump) Organic solvents (Acetone, Toluene, Cyclohexane, Methylene chloride, Chloroform, Carbon tetrachloride)	20–42	Solvent extraction	10	Stable (Chloroform, Methylene chloride, silicone oil, Cyclohexane)	[91]
Bamboo pulp	-	-	135.9	Inorganic oils (pump, liquid paraffin) Organic oils (Sesame) Organic solvents (Chloroform, Heptane, Dimethylacetamide, Paraxylene, sym-Dichloroethane, Petroleum ether, Hexane, Ethanol)	50–150	Extraction, distillation	5	Stable (pump oil, heptane)	[79]
Cellulose from waste board (fluting, box, milkcarton)	MTMS/HDTMS	99.48–99.79	107.2–159	Inorganic oils (Marine diesel, kerosene, gasoline, motor) Organic oils (castor, linseed) Organic solvents (Chloroform, Hexane, Dimethyl sulfoxide, Toluene, Acetone, Ethanol)	65–180	Mechanical compression	30	70–80 (marine diesel oil)	[78]
Cellulose from waste newspaper	-	-	132	Inorganic oils (Gasoline, pump) Organic oils (olive, cooking) Organic solvents (Ethyl acetate, Methylbenzene, Benzene, Acetone, Chloroform, Methanol, Ethanol).	30–52	Distillation (gasoline), Combustion (ethanol), Mechanical compression (pump oil)	5	Stable (gasoline, ethanol) 68.5 (pump oil)	[84]
Chitosan	-	97.67–97.98	-	Inorganic oils (pump, crude, diesel, gasoline) Organic oils (oleic acid) Organic solvents (Ethanol, Acetone, Ethyl acetate, Toluene, Tetraethoxysilane, Ethyleneglycol, Carbon tetrachlorice)	13.11–32.39	Solvent extraction	10	65.97 (diesel)	[85]
Chitosan	MTMS	96.8	152.3	Inorganic oils (vaccuum pump) Organic oils (corn) Organic solvents (Hexane, petroleum ether, Cyclohexane, Toluene, Ethyl acetate, Dichloromethane, Chloroform)	31–63	Mechanical compression	10	Stable (Chloroform, Dichloromethane, vacuum oil, Toluene, Hexane)	[89]
Cotton/Cotton-Cellulose	MTMS	99.4–99.7	130.3–142.8	Inorganic oils (machine)	40–100	Distillation	5	70 (ethanol)	[9]
Cotton-SDS	-	-	-	Inorganic oils (vacuum pump) Organic solvents (Cyclohexane, Ethyl acetate)	145–207	-	-	-	[87]
Lignin-PVA	MTMS	87.54	143	Inorganic oils (kerosene) Organic oils (soybean) Organic solvents (methylbenzene, petroleum ether, n-heptane, trichloromethane)	300–1200	Solvent extraction	10	Stable (toluene)	[90]
Lignin-SA-GO	MTMS	-	144–161	Inorganic oils (pump) Organic solvents (Isooctane, Chloroform, Dichloromethane, n-hexane, Xylene)	5.296–13.214	Mechanical compression	10	39.08 (pump oil) 76.1 (Chloroform)	[24]
Pineapple leaves	MTMS	96.98–98.85	138.6–146.1	Inorganic oils (motor)	26.6–37.9	-	-	-	[88]
Sisal leaves	Cu	99.25–99.76	150.3	Inorganic oils (pump, gasoline) Organic oils (Soybean) Organic solvents (Trichloromethane, Dichloromethane, Dimethyl sulfoxide, Methylbenzene, Cyclohexane, *n*-hexane)	68–165	Distillation	10	40–85 (trichloromethane)	[93]
Sugarcane bagasse	SCCP	92.8–99.2	140.1	Inorganic oils (crude)	17.41–23.08	Solvent extraction	10	70 (crude oil)	[92]
Winter melon	-	>97.5	135	Inorganic oils (pump, crude, diesel, gasoline) Organic oils (corn, sesame, sunflower) Organic solvents (Toluene, Cyclohexane, Hexane, Ethylene glycol, Butyl stereate, Dimethylformamide, Chloroform, Acetone, 2-propanol, Ethanol, Methanol)	15–50	Distillation	5	Stable (ethanol, acetone, gasoline) 48 (crude oil)	[81]

### 4.2. Dye Uptake

The increase in demand for textile products has resulted in about 7 × 10^5^ tons of dye being produced worldwide per year. This direct discharge of untreated dye effluents into the water stream (approximately 2.8 × 105 tons) results in the generation of large textile wastewater, which could inhibit plant growth and present potential toxicity and carcinogenicity [94]. Several technologies have been employed to remove organic dyes from wastewater, such as chemical precipitation and adsorption, electrochemical oxidation and reduction, coagulation and flocculation, membrane separation, H_2_O_2_/ultraviolet (UV), and photocatalysis, among others [95].

In recent times, the search for sustainable, expensive, and reusable technologies has become essential. In this context, biopolymer-based aerogels represent potential candidates for cleaning water bodies due to their low cost, non-toxicity, surface characteristics, and extensive efficiency in dye removal. Table 2 presents examples of biopolymer-based aerogels developed for dye removal, mentioning their most important features.

Chitosan aerogels have been widely employed as an effective alternative for dye rejection due to their electrostatic characteristics [96]. However, sorption performance, stability, and recycling are still critical issues in practical applications, so many authors reported their functionalization with diverse materials. In this context, in 2021, chitosan-based aerogels were combined with soot to evaluate the sorption ability of both anionic and cationic dyes [97]. Three different weight ratios between chitosan and soot were evaluated (1/0, 2/1, and 1/1), showing a low density (from 0.0186 to 0.0342 g/cm^3^) and a high volume of macropores in all cases (>93%). The authors reported that the soot inclusion enhanced both compressive elastic modulus and compressive strength. Furthermore, the sorption isotherms showed that the removal ability of the aerogels for methylene blue, a widely used dye, strongly depended on the soot concentration, suggesting that the carbonaceous particles play a key role in the cationic dyes’ adsorption process.

On the other hand, chitosan concentration is crucial in removing the anionic dye owing to this biopolymer’s protonated amine groups, which allow electrostatic interactions with these colorants. In 2021, a chitosan aerogel enhanced with zeolite powder was developed to obtain an adsorbent composite useful for both anionic and cationic dyes [98]. Pristine and hybrid samples presented densities of 0.022 and 0.043 g/cm^3^, respectively, and the swelling degree of the chitosan aerogel was approximately 25% higher than that of the chitosan-zeolite. Indigo carmine and methylene blue were employed as anionic and cationic dyes to evaluate aerogels’ adsorption capacity. In this context, pristine chitosan aerogel presented high interaction with the anionic dye and low binding ability with the cationic one; meanwhile, the zeolite particles presented the opposite behavior. The adsorption profiles of the hybrid aerogel showed good adsorption ability toward both dyes. These results relate to the combination of the protonated amino groups on the chitosan chains and the zeolite’s negative charge, resulting in an enhanced product.

Similarly, carboxymethyl chitosan aerogel was improved with a magnetic composite, and its efficiency in removing cation and anion dyes was analyzed [99]. The Langmuir model showed that the maximum adsorption capacities for cationic dyes range from 217.43 to 262.27 mg/g (dye/aerogel) and for anionic dyes, from 83.47 to 92.83 mg/g (Figure 6). Additionally, apart from electrostatic interactions, the adsorption mechanism is attributed to hydrogen bonding and hydrophobic and π-π stacking interactions between functional groups and dye molecules. These aerogels showed excellent regeneration after seven adsorption-desorption cycles.

Cellulose aerogels have also been modified with magnetic nanosystems to improve their adsorption properties. In 2020, Payam Arabkhania and Arash Asfaramb developed a three-dimensional aerogel based on bacterial cellulose, graphene oxide, and Fe_3_O_4_ nanoparticles to confer magnetic characteristics [100]. The aerogel was obtained by the freeze-drying technique, and it was characterized by several methods. The hybrid aerogel showed a 3D interconnected porous network structure, a specific surface area of 214.75 m^2^/g, and a total pore volume of 1.135 cm^3^/g. In addition, this superparamagnetic material presented high adsorption efficiency for malachite green dye in both alkaline and acidic conditions. Chemical adsorption was dominant in the adsorption process of this super-light aerogel (6.8 mg cm^−3^), conserving the adsorption efficiency by more than 60% after eight cycles.

Compressibility is a desired property when an aerogel is developed [101]. In 2022, Grishkewich et al. reported elaborating compressible cellulose/(3-mercaptopropyl) trimethoxysilane aerogels by freeze-drying [102]. A sophisticated chemical process was employed to modify the aerogels. The process had an adsorption capacity of 186.7 mg/g and showed a high affinity for methyl orange dye through interactions at the adsorbent surface and physisorption. Other cellulose-based aerogels modified with polydopamine have been particularly interesting because of the multifunctional structure of polydopamine, which increases their hydrophilicity and make them very sensitive to organic dyes. At the same time, they are able to store and convert solar energy even in water that is under an oil layer [103,104].

**Table 2 polymers-15-00262-t002:** Aerogels employed for cationic and anionic dyes rejection.

Raw Material	Second Compound	Porosity (%)	WCA (°)	Dyes	Absorption Capacity (mg/g)	Reusability (Cycles Proved)	Ref.
Cellulose	Chitosan	98.8	-	Methylene blue	780	6	[105]
Biomass (pear fruit)	Graphene	-	-	Crystal violet (CV), methylene blue (MB), rhodamine B (RhB)	57–74	5	[106]
Alginate/gelatin	Graphene oxide	93.5	-	Methylene blue (MB) and Congo red (CR),	196.8–322.6	5	[107]
Cellulose	Polyaniline	-	-	Acid Red G, methylene blue	600.7–1369.6	3	[108]
Cellulose	Zinc Oxide-x	90.64–93.84	109.68	Methyl orange	9000	7	[109]
Cellulose	Polyethylene glycol diglycidyl ether (PEGDE)	93.02	-	Methyl blue	917.43	5	[110]
Pectin	Graphene oxide	-	-	Methyl orange, rhodamine B	419–719	5	[53]
Cellulose (extracted from waste reed) and chitosan.	-	-	-	Congo red	380.23–260.41	6	[111]
Cellulose (coconut)	Polyvinyl alcohol/xanthan gum	96.30–98.32	-	Methylene blue	625	3	[112]
(Ligno)cellulose/cellulose (Wheat Straw)	TEMPO	99.61	-	Methylene blue	5–20	1	[113]
Chitosan	Zeolite	-	-	Indigo carmine, methylene blue	108–221	3	[98]
Cellulose	Polidopamine		143–150	Methylene blue, rhodamine B	11.5–16.5	1	[104]

### 4.3. Pharmaceuticals Recovery

Pharmaceuticals have been widely used worldwide for several decades and are cataloged as emerging pollutants for water since their effect on human health and the environment is unknown [77]. Biopolymeric aerogels have been used to adsorb different kinds of pharmaceuticals that are categorized as antibiotics (such as cycline antibiotics [114], phenicol antibiotics [115]), anti-inflammatories (such as Diclofenac Sodium (DCF) [116]), or analgesics (such as Ibuprofen [117]).

Diverse cycline antibiotics were adsorbed by Li et al. [114] with a carbon aerogel prepared were from glucose and activated with potassium hydroxide. The maximum adsorption capacities reached 1030.5 mg/g for Tetracycline, 3.80 mg/g for Oxytetracycline, 922.93 mg/g for Doxycycline, and 834.92 mg/g for Sulfamethazine. Tetracycline was also adsorbed by Liu et al. [118] using a cellulose aerogel with Zeolitic Imidazolate Framework-67 (ZIF-67) onto polyaniline. Additionally, the aerogel had an adsorption capacity of 409.55 mg/g, maintaining 94% efficiency after 6 cycles. In a further experiment, Wu et al. [119] prepared a cotton cellulose aerogel doped with ZIF-67. By taking advantage of the catalytic sites of ZIF-67, the peroxymonosulfate activates the degradation of tetracycline and p-nitrophenol in 80% of cases within 20 min. The product showed great stability and recyclability and might be used in harsh environments.

The phenicol antibiotics were removed by Liu et al. [115] with a chitosan biochar aerogel. The adsorption capacities reported were 786.1, 751.5, and 691.9 mg/g for chloramphenicol, florfenicol, and thiamphenicol, respectively. The adsorbent continued working after 5 cycles and completely removed the antibiotics within 10 min.

The adsorption of the anti-inflammatory pharmaceutical DCF was evaluated by Feng et al. [116]. They synthesized a cellulose aerogel via microwave-assisted carbonization, adsorbing a maximum of 27.3 mg/g of DCF. A nanocrystalline cellulose aerogel modified with polyvinylamine (PVAm) and reduced graphene oxide (rGO) was fabricated by Lv et al. [120] to adsorb DCF. The result yielded an absorption capacity of 605.87 mg/g. The anti-inflammatory was desorbed with a 0.1 M solution of NaOH, and the adsorbent was reused four times.

The analgesic Ibuprofen was adsorbed by Peydayesh et al. [117] with amyloid fibrils’ aerogels. The result was a maximum adsorption capacity of 69.6 mg/g. Adsorption for beta zone (herbicide) and bisphenol A was also proved, obtaining adsorption capacities of 54.2 and 50.6 mg/g, respectively.

## 5. Perspectives and Conclusions

There is an uncontrolled increase in the contamination of the aquifers that increasingly limits the provision of access to drinking water. Unfortunately, the remaining contaminants can cause predisposition to various chronic diseases. However, biopolymeric fibrous aerogels have exhibited prominent remediation ability against various common pollutants, the production methods are highly cost-effective, and the composition can be very versatile. The trends in the development of new fibrous aerogels include an increase in contaminant specificity in order to enhance remediation efficiency in samples of complex composition. In this sense, studies are still limited to a single pollutant in ideal situations when water pollution comprises multiple compositions in large volume reserves. Consequently, novel biopolymeric fibrous aerogels are also being sought with the capacity to treat or filter larger volumes to facilitate their immersion in real situations and low recirculation cycles to reduce the containment of contaminated water and offer in situ treatment. The changes in hydrophobicity are convenient, but these advanced materials require multifunctionality and proper monitoring of the effects of porosity and performance. Additionally, the reusability capacity is highly suitable, but only for the remediation of new samples of contaminated water and not to compensate for the low effectiveness of initial treatment in the same sample. The chemical composition of the biopolymer has to be highly supervised (e.g., molecular weight and purity) to guarantee adequate reproducibility in obtaining the fibrous aerogel and in water remediation. Finally, the balance between the effectiveness and the cost of the product will determine its possible industrial profitability, an aspect that must be considered to facilitate its application. Hence, after nearly 91 years of research into aerogels, it is still an area of material development with several aspects to be explored and significant and hopeful advances in the face of an emergency for humanity. Water is life, and the search for biopolymeric fibrous aerogels in materials science continues.

## Figures and Tables

**Figure 1 polymers-15-00262-f001:**
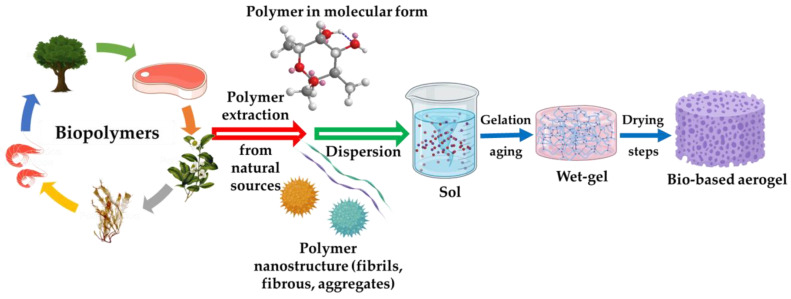
Schematic representation of biopolymeric aerogel elaboration.

**Figure 2 polymers-15-00262-f002:**
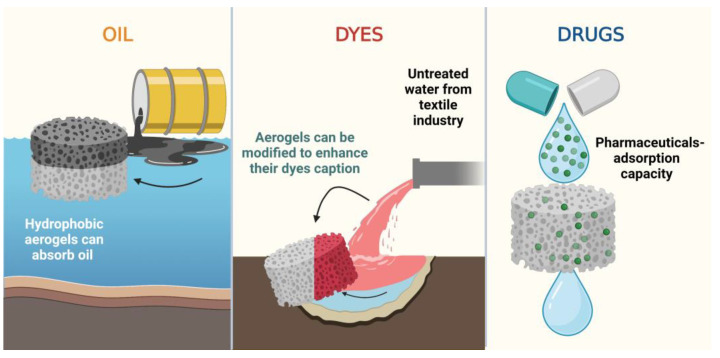
Due to their characteristics, biopolymeric aerogels serve as a removal alternative for oils, dyes, and pharmaceuticals.

**Figure 3 polymers-15-00262-f003:**
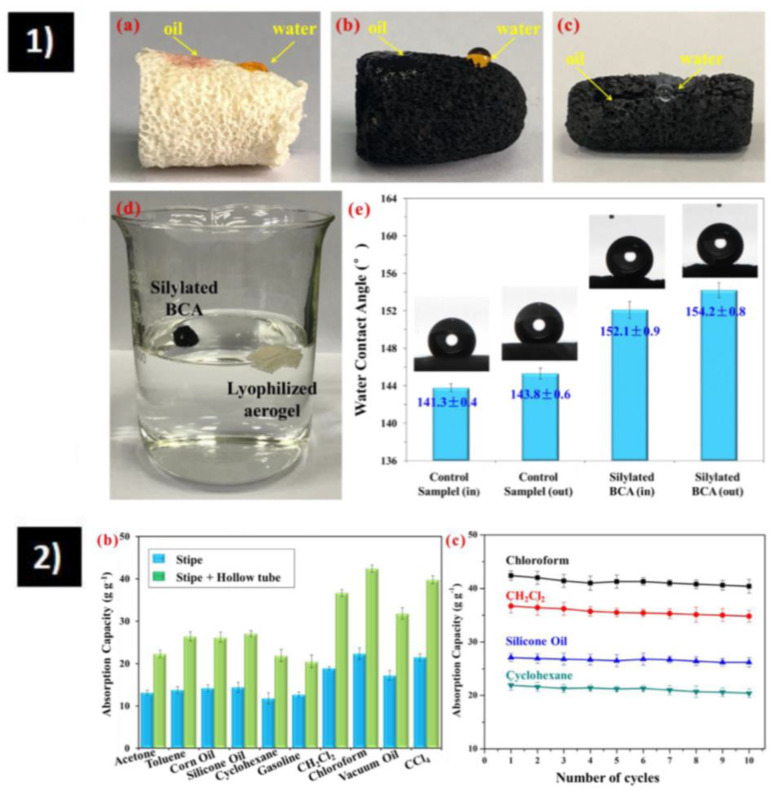
(**1**) Wettability of the bamboo fungus-based aerogel. (**a**) Images illustrating that water and oil droplets are placed on the surface of lyophilized aerogel (above), (**b**) silylated biomass-based carbon aerogel (BCA), and (**c**) freshly cut surface of silylated BCA. (**d**) Image depicting different water repellencies of silylated BCA and lyophilized aerogel. (**e**) Water contact angle on silylated BCA and control sample, wherein the in represents the inner surface and the out indicates the outer surface. (**2**) Oil absorption performance of the bamboo fungus-based aerogel. (**b**) Mass-based absorption capacities of the silylated BCA for various oils and solvents. (**c**) Cyclic absorption performance for the silylated BCA [91].

**Figure 4 polymers-15-00262-f004:**
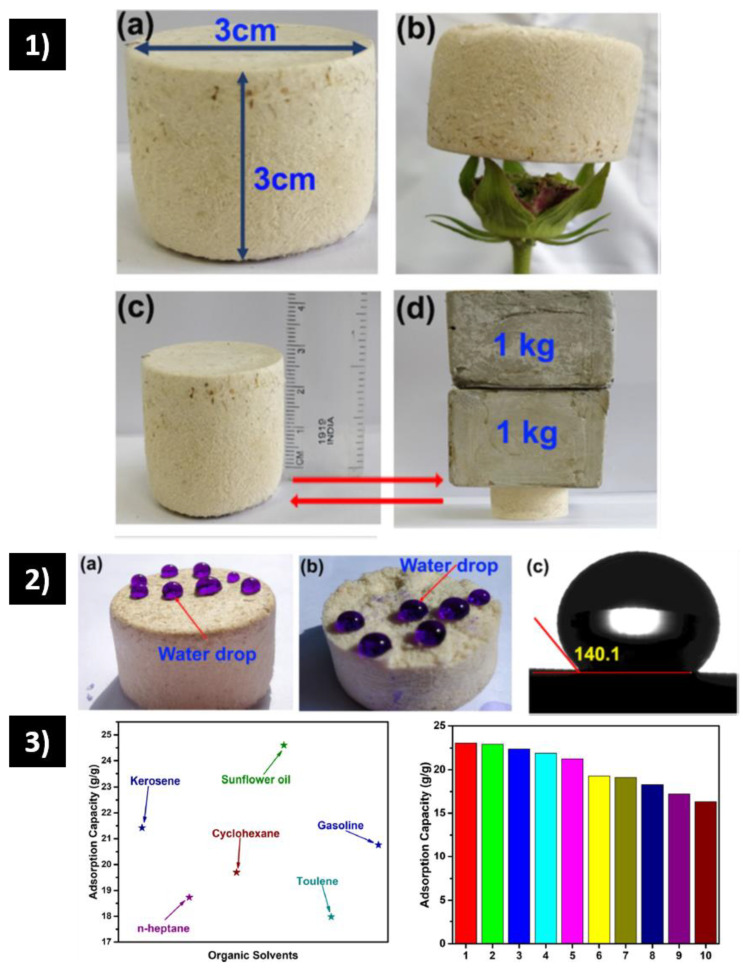
(**1**) Snapshots of bagasse fiber aerogel (**a**) macroscopic structure (**b**) standing on the petals of a flower (**c**) before compression (**d**) supported with 2 kg of weight. (**2**) Water droplets (**a**) on the surface of bagasse fibers aerogel (**b**) water droplets on the cross-sectionally cut of bagasse fibers aerogel (**c**) Water contact angle of bagasse fibers aerogel. (**3**) Various organic solvent adsorption capacities of SBA1. The adsorption capacity of SBA over ten regeneration cycles from the oil-water system [92].

**Figure 5 polymers-15-00262-f005:**
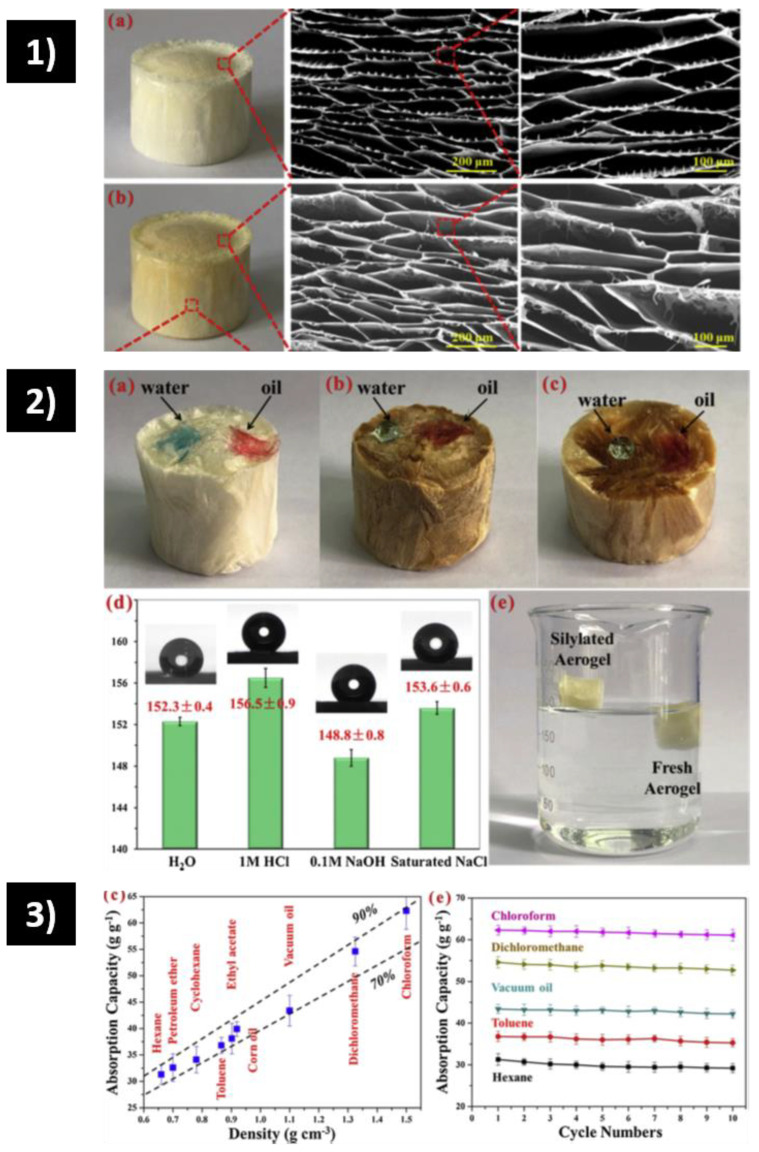
(**1**) Morphology and structure of CS aerogels during different fabrication stages. (**a**) Photograph of pristine aerogel and its cross-sectional SEM images; (**b**) Photograph of MTMS-modified CS aerogel and its longitudinal SEM images. (**2**) Surface wettability of the CS aerogel. (**a**) Photographs of water and oil droplets on the surface of freeze-fried CS aerogel; (**b**) Photographs of water and oil droplets on the MTMS-modified CS aerogel; (**c**) Photographs of water and oil droplets on the freshly cut surface of MTMS-modified CS aerogel; (**d**) Contact angles on the surface of the modified sample for different liquids; (**e**) Images showing the floating of the modified sample in contrast to the sinkage of the fresh sample in the water. (**3**) Oil absorption property of the MTMS-modified CS aerogel. (**c**) The mass-based absorption capacity (g/g) for a wide spectrum of oils and organic solvents. The two dashed lines represent the theoretical volume-based absorption capacity (*v*/*v*) of 90% and 70% for modified aerogel, respectively. (**e**) Cyclic absorption capacities of the modified aerogel [89].

**Figure 6 polymers-15-00262-f006:**
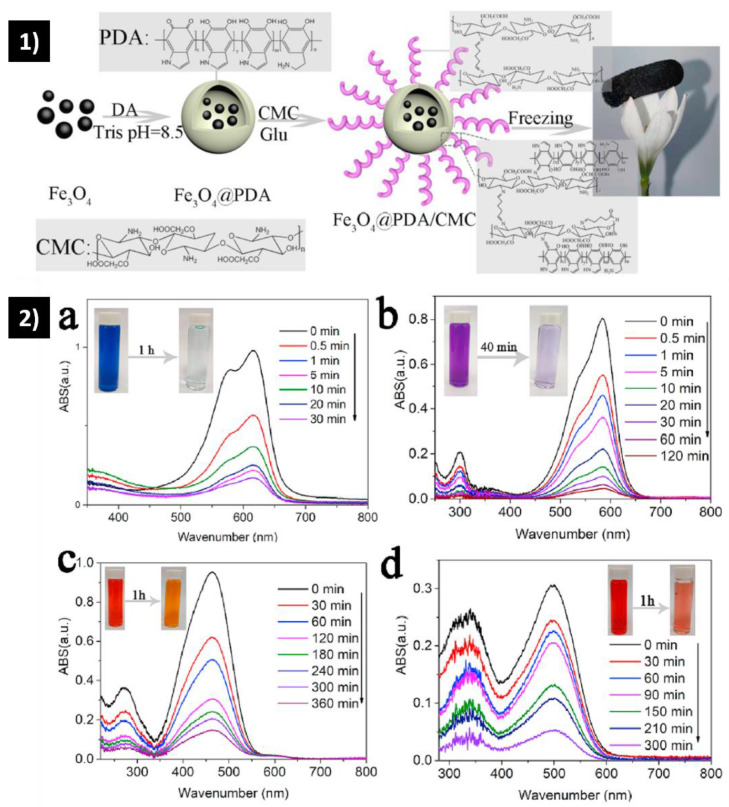
(**1**) Schematic illustration of the preparation of Fe3O4@PDA/CMC aerogel. (**2**) UV spectrum of adsorption times of MB (**a**), CV (**b**), MO (**c**), and CR (**d**) onto Fe3O4@PDA/CMC aerogel [99].

## Data Availability

Not applicable.

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
