# Peer review of "Biopolymeric Fibrous Aerogels: The Sustainable Alternative for Water Remediation"

_polymers, 2023, doi:10.3390/polym15020262_

Round 1

Reviewer 1 Report

This review is devoted to biopolymer fibrous aerogels. The article is relevant and important, since the topic of aerogels has been actively developed in recent years. The article is well written and understandable. The conclusions of the vators are not in doubt. I recommend that the authors improve the following points:

1. It is desirable to state more clearly than this review has many advantages over those already available.

2. Abstract can be expanded a bit.

3. Citation preferred: 10.17516/1998-2836-0271, 10.1039/C2GC36263E, 10.1016/j.indcrop.2012.04.052, 10.3390/foods10112571.

4. The generalized table can be more fully described.

5. In general, the article is written at a good level. In terms of relevance and significance, it corresponds to the subject of the journal. I recommend accepting for publication after minor revisions.

Author Response

This review is devoted to biopolymer fibrous aerogels. The article is relevant and important, since the topic of aerogels has been actively developed in recent years. The article is well written and understandable. The conclusions of the vators are not in doubt. I recommend that the authors improve the following points:

  1. It is desirable to state more clearly than this review has many advantages over those already available.

We agree with the reviewer; the abstract and introduction were rewritten, and now a statement of the novelty of this review is presented.

  1. Abstract can be expanded a bit.

We thank the reviewer for this pertinent commentary. The abstract was expanded in order to attract the readers’ attention.

  1. Citation preferred: 10.17516/1998-2836-0271, 10.1039/C2GC36263E, 10.1016/j.indcrop.2012.04.052, 10.3390/foods10112571.

Following this pertinent observation, references were added to the manuscript.

  1. The generalized table can be more fully described.

We thank the reviewer for this commentary. Tables 1 and 2 are now described in the manuscript.

  1. In general, the article is written at a good level. In terms of relevance and significance, it corresponds to the subject of the journal. I recommend accepting for publication after minor revisions.

Reviewer 2 Report

This manuscript offers a nice review about the biopolymers employed to elaborate aerogels, their synthesis methods, and their applications in water remediation to remove pollutants such as oils, dyes, and pharmaceuticals. And it will be of great help for materials scientists in developing new high-performance materials for water treatment. Therefore, I’d like to recommend the manuscript for publication once the following comments are addressed:

1. In the ‘Introduction’ section, the authors should provide more background on the application of biopolymeric fibrous aerogels in water remediation, which is the main content of this review.

2. The background text behind the leftmost schematic in Figure 1 is unclear.

3. I recommend the authors provide more diagrams to enrich the content of the article. For example, in Section 4.1, it might be better to add more illustrations of employing biopolymers aerogels to remove oil contamination, which can make it easier for readers to understand the article.

4. The authors could add the following references which would again increase the interest to general solar distillation material readers: Science Advances, 2020, 6, eabb4696; ACS Applied Materials & Interfaces, 2021, 13, 7617-7624; Materials Horizons, 2022, 9, 2496-2517.

Author Response

This manuscript offers a nice review about the biopolymers employed to elaborate aerogels, their synthesis methods, and their applications in water remediation to remove pollutants such as oils, dyes, and pharmaceuticals. And it will be of great help for materials scientists in developing new high-performance materials for water treatment. Therefore, I’d like to recommend the manuscript for publication once the following comments are addressed:

  1. In the ‘Introduction’ section, the authors should provide more background on the application of biopolymeric fibrous aerogels in water remediation, which is the main content of this review.

According to the correct reviewer observation, the introduction was improved, and information about the use of biopolymeric aerogels in water remediation was added.

  1. The background text behind the leftmost schematic in Figure 1 is unclear.

We agree with this important commentary, in order to be clearer, Figure 1 was changed.

  1. I recommend the authors provide more diagrams to enrich the content of the article. For example, in Section 4.1, it might be better to add more illustrations of employing biopolymers aerogels to remove oil contamination, which can make it easier for readers to understand the article.

We thank the reviewer for this appropriate observation. Three figures were added to enhance the manuscript quality.

  1. The authors could add the following references which would again increase the interest to general solar distillation material readers: Science Advances, 2020, 6, eabb4696; ACS Applied Materials & Interfaces, 2021, 13, 7617-7624; Materials Horizons, 2022, 9, 2496-2517.

We thank this pertinent observation, and references that enhanced the information presented in the review were added to the manuscript.

Reviewer 3 Report

The manuscript polymers-2113872 reviews the methods of preparation of aerogels and their application, such as water cleaning from pollutants.

I mention even from the beginning that I noticed that the authors confuse the term hydrogel with that of aerogel, related to the preparation methods. The preparation methods presented in the manuscript are characteristic of the hydrogels' preparation and not of aerogels! As for aerogels, they are obtained through a special technique called drying in super-critical conditions. This technique consists in the soaking of a gel containing an organic solvent (such as methanol, ethanol, etc.) under liquid carbon dioxide and the replacing of the liquid in the gel with liquid CO2.

Thus, the entire section dedicated to obtaining aerogels must be removed and written again!

Abstract

- L. 20: “adsorbents for the removal of various pollutants from water” not “water pollutants adsorbents”!

- L. 22: “in the removal of the dyes” not “in removal dyes”!

- The Abstract is ambiguously structured, starting with water pollution, then continuing with the use of aerogels in tissue engineering, then returning to adsorbents for the removal of various pollutants from water. In addition, the presented methods for obtaining of aerogels are not correct! Please review the entire abstract!

1. Introduction

- L. 31-33: “diversity of potential applications they could have because of their versatile chemical structure and the practicality of the methods of obtaining them”? The sentence is ambiguous! Please revise it!

- The introduction is poor in information and very general!

- The polymers that are used in making aerogels are not discussed, and moreover, it is not mentioned whether this review presents information about aerogels based on natural or synthetic polymers! Natural polymers are mentioned in the Abstract, but this paragraph is completely missing in Introduction section!

- There are not presented the latest news in the field of aerogels and why the authors chose to discuss in this review only about water bioremediation!

- There is no information regarding the use of aerogels as water bioremediation, but only one reference!

- There is no information related to the novelty of this manuscript!

- Please add all this information in Introduction section and specify the novelty of this review by comparing it with other similar information from literature!

2. Biopolymers employed

- L. 61: “proteins and nucleic acids”? Where did the authors find information about the obtaining of aerogels from proteins and nucleic acids or about the usage of nucleic acids in wastewater treatment?

- Natural polymers are presented very briefly, without bringing important information to the reader's attention! Each polymer should be presented in an independent paragraph, accompanied by its important characteristics. Moreover, it must be shown how each type of natural polymer used in the preparation of aerogels influences both the structure of the aerogels and their properties. The experimental data for each type of aerogel must be mentioned and discussed among themselves!

3. Methods of obtention

- L. 105: “Methods of obtention”? This is not a proper name for a title! The authors can use “Preparation methods!

- This paragraph must be completely deleted and the correct information about the obtaining of aerogels must be added!

4. Aerogels application in water remediation

4.1 Oil recovery

- L. 209: What is WCA? Please add the explanation for all the abbreviations used in this manuscript!

- L. 209-212: “To increase hydrophobicity the surface of biopolymeric aerogels might be functionalized with methods such as a silylation routine”! There is no information how was improved the aerogels’ characteristics by using different methods! Please add and compare the experimental data from this article, evidencing each method!

- More experimental data must be added to each example from this section!

4.2 Dyes uptake

- L. 359: In Table 2, adding a second compound does not represent a Modification! The modification leads you to think of a derivatization of the polymer. In my opinion, "Modification" should be changed to "Second compound" or another variant that indicates the addition of a second compound in the network.

- Each example from this section must be improved with experimental data related to the porosity, density, absorption capacity, etc., in order to evidence the characteristics and improved properties of the aerogels!

In conclusion, the paper needs a lot of improvements, must be rigorously checked related to the terminology and the authors must bring more information related to the field of aerogels, with more detailed explanations.

Author Response

Abstract

- L. 20: “adsorbents for the removal of various pollutants from water” not “water pollutants adsorbents”!

We agree with this pertinent commentary and we corrected the mistake.

- L. 22: “in the removal of the dyes” not “in removal dyes”!

We are sorry for this mistake, we corrected the manuscript.

- The Abstract is ambiguously structured, starting with water pollution, then continuing with the use of aerogels in tissue engineering, then returning to adsorbents for the removal of various pollutants from water. In addition, the presented methods for obtaining of aerogels are not correct! Please review the entire abstract!

Following these pertinent observations, we rewrote the abstract and corrected the information.

  1. Introduction
    - L. 31-33: “diversity of potential applications they could have because of their versatile chemical structure and the practicality of the methods of obtaining them”? The sentence is ambiguous! Please revise it!

- The introduction is poor in information and very general!

- The polymers that are used in making aerogels are not discussed, and moreover, it is not mentioned whether this review presents information about aerogels based on natural or synthetic polymers! Natural polymers are mentioned in the Abstract, but this paragraph is completely missing in Introduction section!

- There are not presented the latest news in the field of aerogels and why the authors chose to discuss in this review only about water bioremediation!

- There is no information regarding the use of aerogels as water bioremediation, but only one reference!

– There is no information related to the novelty of this manuscript! - Please add all this information in Introduction section and specify the novelty of this review by comparing it with other similar information from literature!

Based on these pertinent commentaries, the introduction section was rewritten in order to show the importance of finding new alternatives for water remediation since water pollution is one of the most relevant current ecological problems.  The text was enhanced with information about biopolymers employed for the aerogels elaboration and enriched with current references to water remediation. Furthermore, we emphasize the novelty of this work.

  1. Biopolymers employed

- L. 61: “proteins and nucleic acids”? Where did the authors find information about the obtaining of aerogels from proteins and nucleic acids or about the usage of nucleic acids in wastewater treatment?

We apologize for this mistake, the word “nucleic acids” were erased, and references supporting the information of proteins were added.

* Peydayesh, M.; Vogt, J.; Chen, X.; Zhou, J.; Donat, F.; Bagnani, M.; Müller, C.R.; Mezzenga, R. Amyloid-Based Carbon Aerogels for Water Purification. Chemical Engineering Journal 2022, 449, 137703.

* Ozden, S.; Monti, S.; Tozzini, V.; Dutta, N.S.; Gili, S.; Caggiano, N.; Link, A.J.; Pugno, N.M.; Higgins, J.; Priestley, R.D. Egg Protein Derived Ultralightweight Hybrid Monolithic Aerogel for Water Purification. Materials Today 2022, 59, 46–55.

- Natural polymers are presented very briefly, without bringing important information to the reader's attention! Each polymer should be presented in an independent paragraph, accompanied by its important characteristics. Moreover, it must be shown how each type of natural polymer used in the preparation of aerogels influences both the structure of the aerogels and their properties. The experimental data for each type of aerogel must be mentioned and discussed among themselves!

Following this timely observation, each biopolymer is presented in an independent paragraph, showing its most important properties.

  1. Methods of obtention

- L. 105: “Methods of obtention”? This is not a proper name for a title! The authors can use “Preparation methods!

According to the correct observation of the reviewer, we changed the section title to “Preparation methods”.

- This paragraph must be completely deleted and the correct information about the obtaining of aerogels must be added!

Section 3 was rewritten in order to present the pertinent information.

  1. Aerogels application in water remediation

4.1 Oil recovery

- L. 209: What is WCA? Please add the explanation for all the abbreviations used in this manuscript!

We are sorry for this misunderstanding, the WCA acronym is now defined in the manuscript.

- L. 209-212: “To increase hydrophobicity the surface of biopolymeric aerogels might be functionalized with methods such as a silylation routine”! There is no information how was improved the aerogels’ characteristics by using different methods! Please add and compare the experimental data from this article, evidencing each method!

We agree with this pertinent commentary. The chemistry behind the silylation is now described, and the characteristics of the pristine and modified aerogels are now compared and discussed.

- More experimental data must be added to each example from this section!

In order to attempt this timely observation, we added information about experimental results in the whole of section 4.

4.2 Dyes uptake

- L. 359: In Table 2, adding a second compound does not represent a Modification! The modification leads you to think of a derivatization of the polymer. In my opinion, "Modification" should be changed to "Second compound" or another variant that indicates the addition of a second compound in the network.

We agree with this commentary, “modification” was changed to “second compound” in order to avoid confusion.

- Each example from this section must be improved with experimental data related to the porosity, density, absorption capacity, etc., in order to evidence the characteristics and improved properties of the aerogels!

To attempt this timely observation, information on each example was added to have a deeper evaluation of the aerogels for dye removal.

Round 2

Reviewer 3 Report

The authors paid attention to all comments, thus, I agree with the publication of the present version, after a minor improvement:

-        At section 2 „Biopolymers employed”, the sub-chapter „Other” must be numbered like the other sub-chapters and thus, I would propose the title „2.4. Miscellaneous biopolymers”.